# A 2-Dimensional State Space Layer for Spatial Inductive Bias

**Ethan Baron**\*, **Itamar Zimerman**\* **& Lior Wolf**
The Blavatnik School of Computer Science, Tel Aviv University
`{barone,zimerman1}@mail.tau.ac.il`
`wolf@cs.tau.ac.il`

## Abstract

A central objective in computer vision is to design models with appropriate 2-D inductive bias. Desiderata for 2-D inductive bias include two-dimensional position awareness, dynamic spatial locality, and translation and permutation invariance. To address these goals, we leverage an expressive variation of the multidimensional State Space Model (SSM). Our approach introduces efficient parameterization, accelerated computation, and a suitable normalization scheme. Empirically, we observe that incorporating our layer at the beginning of each transformer block of Vision Transformers (ViT), as well as when replacing the Conv2D filters of ConvNeXT with our proposed layers significantly enhances performance for multiple backbones and across multiple datasets. The new layer is effective even with a negligible amount of additional parameters and inference time. Ablation studies and visualizations demonstrate that the layer has a strong 2-D inductive bias. For example, vision transformers equipped with our layer exhibit effective performance even without positional encoding. Our code is available at this git https URL.

## 1 Introduction

Incorporating image-specific inductive bias into computer vision networks could play a crucial role in their success, by shaping the hypothesis space in a way that fits image data and improves generalization. Common ingredients of image-specific inductive bias include two-dimensional neighborhood structure, locality, translation equivariance and invariance, and extraction of hierarchical features. Traditionally, it was injected into the model through the backbone architecture. However, more recently, it has been modeled as part of the data. For example, two-dimensional neighborhood structures are typically expressed in one of two ways: (i) Vision Transformers (Dosovitskiy et al., 2021) use 1-D positional encoding (Vaswani et al., 2017), which is considered weak inductive bias. (ii) ConvNets employ 2-D kernels, which provide strong priors on the underlying image structure (Ulyanov et al., 2018).

Most ConvNets employ relatively small filters in the convolution layers, and the balance between local and global features is handled by increasing the receptive field with depth. However, other kernel sizes can be beneficial. For example, ConvNeXt improved ResNet by $0.7\%$ on Imagenet, by only increasing its kernel size from $3 \times 3$ to $7 \times 7$ (Liu et al., 2022). More generally, using fixed-size filters limits the type of dependencies the layer can capture.

The objective of this study is to develop a new layer that is adept at integrating both local and global spatial features, with a particular focus on a two-dimensional neighborhood structure. We accomplish this by building on recent developments in 1-D SSM-based layers, which are renowned for capturing various types of dependencies. By extending this 1-D concept to 2-D, our layer is deeply rooted in control theory, and much like its predecessors, maintains a strong bias towards position awareness and parameter efficiency.

**Our main contribution** is the 2D-SSM layer, which is a new spatial layer based on Roesser's model for multidimensional state space (Kung et al., 1977). We show that simple design choices,

---

\*These authors contributed equally to this work.

such as diagonalization and normalization, can make the layer numerically stable and efficiently computed without recurrence using a 2-D convolution (left panel of Fig. 1). Our layer has some unique properties, including: (i) A strong inductive bias towards two-dimensional neighborhood and locality, which stems from the multi-dimensional recurrent rule. As far as we know, this novel concept does not appear in other layers, (ii) The new layer can capture unrestricted controllable context. The SSM parameters of the layer can be focused on short or long, horizontal, vertical, or diagonal dependencies (middle panel of Fig. 1). (iii) The layer is parameter-efficient and can express kernels of any length via 8 scalars. Visualization and ablation studies demonstrate these key aspects of our layers. Finally, the layer is well-grounded in control theory, and further theoretical analysis shows that it generalizes S4ND (Nguyen et al., 2022) and proves its greater expressiveness.

Empirically, we show that our layer can be used as a general-purpose booster for vision transformers (the schematic architecture is illustrated in the right panel of Fig. 1), with negligible additional parameters and computation at inference. Furthermore, it appears that our 2D-SSM surpasses standard methods, such as incorporating positional encoding, in effectively integrating positional bias into Vision Transformers. Finally, we demonstrate that incorporating the new layer instead of 2D convolutions, within the state-of-the-art CNN architecture ConvNeXt Liu et al. (2022), improves its accuracy, without significantly changing the number of parameters.

## 2 BACKGROUND AND NOTATIONS

**Framing** Our research delves into two emerging research domains. The first domain focuses on the development of multi-axes **global** convolution techniques. Although 1-D (long) global convolution has shown promise in 1-D sequence modeling, leveraging methods such as SSM (Dao et al., 2022; Gu et al., 2021a;b; Gupta, 2022; Mehta et al., 2022) or other recent approaches (Fu et al., 2023),(Poli et al., 2023),(Li et al., 2022b), its applicability and effectiveness in modern computer vision tasks remain uncertain. Our work aims to explore and highlight the potential of these techniques in this domain, by extending them into 2-D.

The second domain investigates the synergistic combination of attention and SSM in 1-D modeling across various domains(Ma et al., 2022; Saon et al., 2023; Islam et al., 2022; Zuo et al., 2022; Dao et al., 2022; Mehta et al., 2022). For example, the SSM-based H3 (Dao et al., 2022) outperforms GPT-Neo-2.7B (Black et al., 2021) (as well as other transformers of the same size) with only 2 attention layers. However, the question of whether these components are complementarity in 2-D modeling remains unanswered. We provide empirical evidence supporting the complementary nature of these components.

**State Space Model (SSM)** The state space model maps an input scalar function $u(t) : \mathbb{R} \to \mathbb{R}$ to a N-D latent state $x(t) \in \mathbb{R}^N$ before projecting it to an output signal $y(t) : \mathbb{R} \to \mathbb{R}$:

$$\dot{x}(t) = Ax(t) + Bu(t), \quad y(t) = Cx(t) + Du(t) \tag{1}$$

The use of SSMs is widespread across numerous scientific disciplines and is closely associated with latent state models, such as Hidden Markov Models. There is a well-known connection between linear time-invariant SSMs, such as 1 and continuous convolution, thus allowing efficient training using the aforementioned equation as a discrete convolution. The S4 (Gu et al., 2021a) and LSSL (Gu et al., 2021b) layers leveraged the SSM as a black-box representation in deep sequence modeling, with learned parameters A, B, C, and D, and achieved strong results on several tasks, especially ones that require handling long-range dependencies, such as the Long Range Arena (LRA) (Tay et al., 2020), audio generation (Goel et al., 2022), and long-text processing (Mehta et al., 2022; Golub & He, 2016; Dao et al., 2022).

The underlying reasons for the suitability of S4 and SSMs for modeling long sequences were recently analyzed (Li et al., 2022b; Fu et al., 2023). It was found that (i) employing global kernels with decaying structures, and (ii) using regularized kernels, are both critical design choices in this area.

**Multi-dimensional State Space Layers** As far as we know, S4ND (Nguyen et al., 2022) is the only previous SSM-based layer that can naturally handle multidimensional data. S4ND is built on top of S4, and contains $M$ separate instances of S4, where $M$ is the number of axes. On the forward path, each S4 layer, which we denote as $SSM_g$ factors a one-dimensional local kernel $k_g$, and a global kernel $K$ is computed as the outer products of those local kernels.

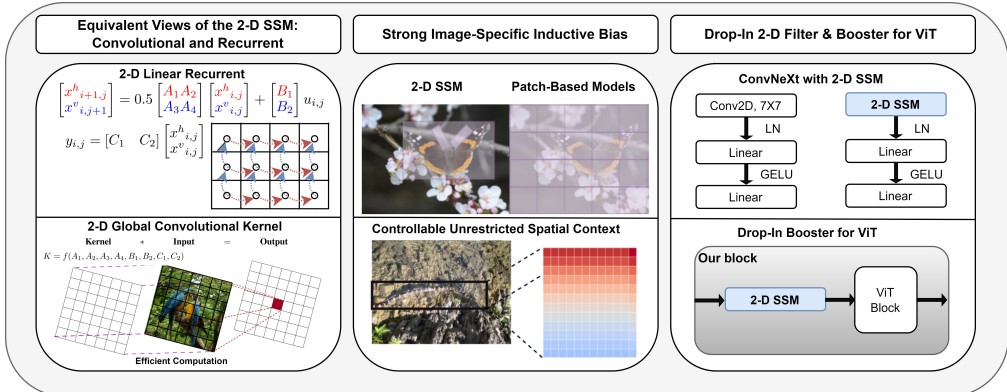

Figure 1: (Left) The 2-D SSM layer is parameterized by A, B, C, and D. It is built on top of a two-axis linear recurrent and can be efficiently computed using 2-D convolution. (Center) Since the layer is based on two-dimensional recurrence, it exhibits a strong bias toward positional awareness. The recurrent is unrestricted, allowing the layer to operate on 2-D sequences of any length. The values of $A_1, A_2, A_3$, and $A_4$ control the layer's focus, enabling it to capture short or long spatial dependencies in horizontal, vertical, or diagonal directions, as opposed to patch-based models. (Right) The layer can be easily integrated into ViT by applying it to the two-dimensional sequence of patches at the beginning of each transformer block, or by replacing the standard Conv2D in ConvNeXt.

**Roesser's 2D-State Space Model** The attempt to extend the classical SSM for 2-D multi-axes systems was thoroughly studied in the past. (Kung et al., 1977; Eising, 1978; Fornasini & Marchesini, 1978; Kurek, 1985; Givone & Roesser, 1972; Hinamoto, 1980) notes a few different formulations of the problem. We employ Roesser's SSM model (Kung et al., 1977) as our discrete multi-axial model, which is the most general form of 2-axial and N-axial state-space models. As opposed to other SSMs already in use in machine learning, this SSM uses $M$ states, one per axis, rather than just one. The model in 2-D form is presented here:

$$x_{i,j} = \begin{bmatrix} x^h{}_{i,j} \\ x^v{}_{i,j} \end{bmatrix}, \quad y_{i,j} = \begin{bmatrix} C_1 & C_2 \end{bmatrix} \begin{bmatrix} x^h{}_{i,j} \\ x^v{}_{i,j} \end{bmatrix}, \quad \begin{bmatrix} x^h{}_{i,j+1} \\ x^v{}_{i+1,j} \end{bmatrix} = \begin{bmatrix} A_1 A_2 \\ A_3 A_4 \end{bmatrix} \begin{bmatrix} x^h{}_{i,j} \\ x^v{}_{i,j} \end{bmatrix} + \begin{bmatrix} B_1 \\ B_2 \end{bmatrix} u_{i,j} \quad (2)$$

where the state $x_{i,j} \in R^{2N}$ is the concatenation of the horizontal $x^v{}_{i,j} \in R^N$ and vertical $x^h{}_{i,j} \in R^N$ states, the system matrices are $A_1, A_2, A_3, A_4 \in R^{N \times N}$ and the input and output matrices are $B_1, B_2, C_1, C_2 \in R^N$. There is also a learned parameter D that behaves as a skip-connection, omitted from now on for brevity.

Additional related work is presented in Appendix I.

## 2.1 NOTATION

The notation follows as closely as possible the notation used in the state-space layer literature (Gu et al., 2021a; Gupta, 2022; Gu et al., 2022). Specifically, we use $H$ as the number of channels, $N$ as the state's hidden dimension, $L$ as the sequence length, $n_{ssm}$ as the number of non-shared channels, and treat $A, B, C, D \in \mathbb{R}$ as the system matrices. Note that for brevity the system matrices are treated as real-valued, even though we also test a complex version of them. The number of axes is set to $M$.

Although $N$-D SSM can be used in an $N$-Dimensional manner, since our paper focuses on using this model as a regularization method for ViT backbones and for simplifying the reading experience, we will treat it as a 2-D SSM.

$L_i \in \mathbb{R}$ is the sequence length along the $i$ axes, $L_{tot} = L_1 \cdot L_2$ is the total signal size, and $L_{max} = \max(L_1, L_2)$.

The kernel notation introduces $K$, which is the computed 2-D kernel such that $Y = U * K$, where $*$ denotes discrete convolution.

$$y_{i,j} = C_1 x^h{}_{i,j} + C_2 x^v{}_{i,j} = \sum_{0 \leq \hat{i} \leq i} \sum_{0 \leq \hat{j} \leq j} \left( C_1 k^h{}_{\bar{i},\hat{j}} + C_2 k^v{}_{\bar{i},\hat{j}} \right) u_{\hat{i},\hat{j}} \quad (3)$$

## 3 METHOD

In this section, we present the core of the 2-D SSM layer, which is our main technical contribution. This layer maps a 2-dimensional sequence $u_{i,j}$ to $y_{i,j}$, where $u_{i,j}, y_{i,j} \in \mathbb{R}$ for all $0 \leq i \leq L_1$ and $0 \leq j \leq L_2$. Similarly to previous SSM-based layers, we extend the core of our layer to a multi-directional and multidimensional layer, detailed in Appendix D.

### 3.1 2-D RECURRENT STATE SPACE MODEL AS CONVOLUTION

Eq. 2 is defined in a recursive manner. However, for reasons of efficiency, it is best to define operators in closed form and in a way that is concurrent with parallel processing. Inspired by previous work (Gu et al., 2021a; Gupta, 2022; Ma et al., 2022; Gu et al., 2022), we exploit the fact that the SSM is linear and can therefore be expressed as a convolution with a kernel $K$. To do so, we first unroll the recurrent rule and then describe it in a closed-form formulation.

For simplicity, we assume that the initial states are zeros $\forall j \geq 0 : x^h{}_{-1,j} = 0$, and $\forall i \geq 0 : x^v{}_{i,-1} = 0$. The horizontal and vertical states at $i = 0, j = 0$ are:

$$x^h{}_{0,0} = B_1 u_{0,0}, \quad x^v{}_{0,0} = B_2 u_{0,0} \tag{4}$$

By applying the recurrent rule once at each axis:

$$x^h{}_{1,0} = A_1 B_1 u_{0,0} + A_2 B_2 u_{0,0} + B_1 u_{1,0}, \quad x^v{}_{0,1} = A_3 B_1 u_{0,0} + A_4 B_2 u_{0,0} + B_2 u_{0,1} \tag{5}$$

$$x^v{}_{1,0} = B_2 u_{1,0}, \quad x^h{}_{0,1} = B_1 u_{0,1} \tag{6}$$

Next, we compute $x^h{}_{1,1}$ given $x^h{}_{0,1}$ and $x^v{}_{0,1}$.

$$x^h{}_{1,1} = A_1 A_3 B_1 u_{0,0} + A_1 A_4 B_2 u_{0,0} + A_1 B_2 u_{0,1} + A_2 B_1 u_{0,1} + B_1 u_{1,1}, \tag{7}$$

$$= k^h_{1,1} u_{0,0} + k^h_{1,0} u_{0,1} + k^h_{0,0} u_{1,1} \tag{8}$$

and in general

$$x^h{}_{i,j} = \sum_{0 \leq \hat{i} \leq i} \sum_{0 \leq \hat{j} \leq j} k^h_{\bar{i},\hat{j}} u_{\hat{i},\hat{j}} \quad , x^v{}_{i,j} = \sum_{0 \leq \hat{i} \leq i} \sum_{0 \leq \hat{j} \leq j} k^v_{\bar{i},\hat{j}} u_{\hat{i},\hat{j}} , \tag{9}$$

where as explained in the notation, each element $k^h_{\hat{i},\hat{j}}$, $k^v_{\hat{i},\hat{j}}$ is an aggregation of $A_1, A_2, A_3, A_4, B_1, B_2$ multiplications (e.g Eq. 7) and is associated with a single path from co-ordinate $(0,0)$ to $(i,j)$, as presented in Fig 2.

By plugging Eq. 9 in Eq. 2 one obtains:

$$y_{i,j} = C_1 x^h{}_{i,j} + C_2 x^v{}_{i,j} = \sum_{0 \leq \hat{i} \leq i} \sum_{0 \leq \hat{j} \leq j} \left( C_1 k^h_{\bar{i},\hat{j}} + C_2 k^v_{\bar{i},\hat{j}} \right) u_{\hat{i},\hat{j}} \tag{10}$$

and the convolutional kernel $K$ is

$$\forall i,j : K_{i,j} = C_1 k^h_{\bar{i},\hat{j}} + C_2 k^v_{\bar{i},\hat{j}} \tag{11}$$

### 3.2 EFFICIENT AND STABLE PARAMETERIZATION

**Parameter diagonalization** Computing $K$ is difficult for two reasons. Firstly, the number of elements in each $k_{i,j}$ is exponential in $i$ and $j$, since it is equivalent to the number of paths from $(0,0)$ to $(i,j)$. Secondly, calculating the powers of non-diagonal matrices $A_1, A_2, A_3, A_4$ becomes costly when $L_1, L_2$ are large. To overcome these challenges, we parameterized the system matrices $A_1, A_2, A_3, A_4$ as diagonal. This allows for efficient summation of at most $2L_{max}$ elements in $k^h_{i,j}$. Although diagonalization limits the expressiveness of our SSM, previous works have shown its effectiveness in one-dimensional cases (Gu et al., 2022; Gupta, 2022; Mehta et al., 2022; Gupta et al., 2022).

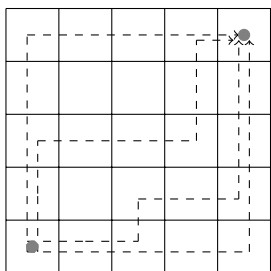

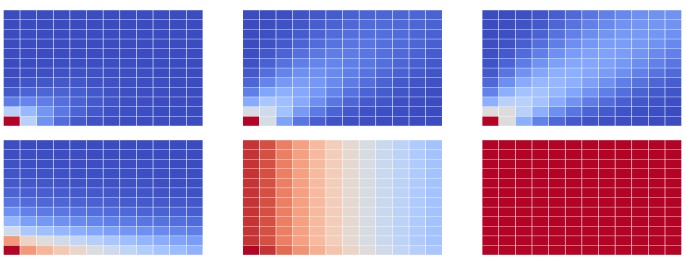

Figure 3: The kernels before and after the modifications of Sec. 3.2. Each column is created by the same $A_1...A_4, B_1, B_2, C_1, C_2 \in \mathbb{R}$ parameters. The first row is the normalized 2-D SSM formulation explained in 2, the second is the outcome of Eq. 12 and performing Eq. 13, which is the kernel formulation we use. The bottom left corner of each heatmap is $K_{0,0}$. The figures demonstrate that before the relaxation, the kernels displayed a diagonal tendency while afterward, they exhibited a more diverse and versatile pattern.

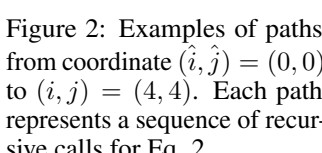

Figure 2: Examples of paths from coordinate $(\hat{i}, \hat{j}) = (0,0)$ to $(i,j) = (4,4)$. Each path represents a sequence of recursive calls for Eq. 2.

**Limiting the parameters**   $A_i \in \mathbb{R}^{N \times N}$ is now a diagonal matrix. We'll denote the eigenvalues of $A_i$ by $(\alpha_1, \alpha_2...\alpha_N) := \mathbf{diag}(A_i)$. Each $\alpha_i$ value behaves separately until the computation of K (Eq. 11). Thus, for $\alpha_i > 1$, $\lim_{z \to \infty} \alpha^z = \infty$. Therefore, we limit $\alpha_i \in [0,1]$ by parameterized it by $\alpha_i := sigmoid(\hat{\alpha_i})$ and optimizing $\hat{\alpha_i}$ instead.

**Normalization**   There is a scaling problem that arises from Eq. 7. Since $k^h_{1,1} = A_1 A_3 B_1 + A_1 A_4 B_2$, even if $A_1 A_3 B_1, A_1 A_4 B_2 \leq 1$ ,$k^h_{1,1}$ can be greater than 1. The same behavior makes the kernel explode.

We would like to keep $\forall i, j : 0 \leq k^h_{i,j} \leq 1$ and thus we employ a straightforward normalization mechanism, in which we divide by two each time we compute Eq. 2. This Eq. is thus replaced by:

$$\begin{bmatrix} x^h_{i+1,j} \\ x^v_{i,j+1} \end{bmatrix} = 0.5 \begin{bmatrix} A_1 A_2 \\ A_3 A_4 \end{bmatrix} \begin{bmatrix} x^h_{i,j} \\ x^v_{i,j} \end{bmatrix} + \begin{bmatrix} B_1 \\ B_2 \end{bmatrix} u_{i,j} \tag{12}$$

**Relaxation of the kernel**   When imposing a "divide by two" normalization for every $k_{i,j}$, we impose a kernel formulation that is much more biased towards modeling diagonal kernels, i.e., if $|i - j|$ is small, $k^h_{i,j}$ is much larger than when $|i - j|$ is large.

Thus, we relax the normalization in the first row and column as follows. When calculating $k^h_{i,0}, k^h_{0,j}$ we use Eq. 2. Additionally, for $K_{i,0}, K_{0,j}$, we use $\hat{C}_1 = 2C_1, \hat{C}_2 = 2C_2$ in the following manner:

$$K_{0,j} = \hat{C}_1 k^h_{0,j} + \hat{C}_2 k^v_{0,j} \tag{13}$$

Figure 3 illustrates examples of different kernels before and after the relaxation.

We note that these modifications are straightforward and probably not optimal. The development of optimal normalization according to the kernel formulation is an interesting direction for future work.

### 3.3   COMPUTATION AND COMPLEXITY

**Training Complexity**   The training process has two parts. First, we calculate the kernel K. The calculation itself is explained thoroughly in Appendix B and has a time complexity of $O(L_{tot} L_{max} N)$, which is not dependent on B, and it is much faster than naive computation thanks to several design choices, such as parameters diagonalization, pre-processing, and a sophisticated caching procedure.

Next, we apply a convolution between $K$ and $U$, using the classical procedure of FFT, element-wise multiplication, and inverse FFT. The complexity of this step is $O(B L_{tot} \log(L_{tot}))$ where $B$ is the batch size. Hence, the total complexity is $O(L_{max} N L + B \log(L_{tot}) L_{tot})$

**Inference Complexity** During inference, the 2-D SSM can pre-compute the convolution kernels, resulting in an additional computation cost of only the 2-dimensional convolution of the layer input and the signal, which takes $L_{tot} \log(L_{tot})$ operations. Therefore, the additional computation overhead relative to the vanilla ViT is minimal, as the quadratic complexity dominates the overall complexity. For an empirical analysis of the time and memory complexities, we refer the reader to Appendix H.

### 3.4 Complex and Real SSMs

While most SSM-based deep learning models, e.g S4 (Gu et al., 2021a) or DLR (Gupta et al., 2022), use a complex SSM, MEGA used EMA, which can be interpreted as a restriction of the diagonal SSM to real numbers. Our 2-D SSM layer can be built over real or complex parametrization. The complex-SSM based model that we examined is described in detail in Appendix E.1.

## 4 Model Analysis

We study the expressiveness of the 2-D SSM layer, as well as its unique spatial inductive bias.

### 4.1 Expressiveness

We compare the expressiveness of our 2-D SSM layer with S4ND (Nguyen et al., 2022), a very recent layer that is also based on multidimensional multi-axes SSM. We first introduce the key differences between our layer and S4ND, and then demonstrate the expressiveness gap.

**The relationship between 2-D SSM and S4ND** The main difference between S4ND and 2-D SSM is that S4ND runs a standard 1-D SSM over each axis independently, and those functions are combined to form a global kernel. In contrast, our model learns multi-dimensional functions over multi-axes data directly. This difference arises from the fact that the 2-D SSM has additional system matrices, $A_2, A_3$, which aggregate and process information from different axes.

**2D-SSM is a generalization of S4ND** When restricted to 2-dimensional space, given a 2-D SSM model, S4ND is obtained by restricting $A_2, A_3$ to be zeros, and setting $A_1, A_4$ as the system matrices of S4ND. Additionally, to replicate S4ND in 2D-SSM, one should initialize the states of the 2D-SSM by the kernels factorized from S4ND.

**Tensor rank as a criterion for expressiveness** Over the years, several criteria were proposed for measuring the expressiveness of neural and classical models, such as VC-dimension (Vapnik & Chervonenkis, 2015), norms, and Rademacher complexity (Bartlett & Mendelson, 2002). Inspired by (Cohen et al., 2016), we employ tensor rank as our measure, and prove the followings theorems:

**Theorem 4.1.** *The $8$ parameters of the $2$-D SSM can express full-rank kernels*

**Theorem 4.2.** *S4ND can express kernels of rank $1$ only.*

**Assumptions** For simplicity, we assume that both the 2-D SSM and S4ND layers contain one channel and one hidden dimension. In this case, the SSM matrices are scalars. When the assumption about the number of channels is omitted, the rank of kernels that S4ND can express increases from $1$ to $N$. However, this comes with the cost of $MNr$ additional parameters, where $M$ is the number of axes and $r$ is the required rank for S4ND. It should be noted that the authors of S4ND did not evaluate the performance of models with $r > 1$.

Under these assumptions, the proof of Theorem 4.1 is specified in Appendix C.1. The proof of 4.2 is trivial, and derives from the fact that to compute a global multi-axis kernel $K$, S4ND takes the outer product operation on the per-axis kernels $k_m \in \mathbf{C}^{L_m \times 1}$ for all $m \in [M]$. Since each kernel is a vector, it is clear that:

$$\mathbf{rank}(K) = \mathbf{rank}(k_1 \otimes k_2 \otimes \ldots k_M) = 1 \tag{14}$$

### 4.2 Image-Specific Inductive Bias

**Two-dimensional Position-Awareness** Our layer is grounded by a two-dimensional linear recurrent (Eq. 2 ,Fig. 1,left), as a result, positional information is taken into account by design when the kernel K is computed from the parameters $A, B, C$, and $D$. This is a unique property without a counterpart in other modern layers. Furthermore, as can be seen in Sec. 5, our method is highly effective in inserting positional bias into the transformer, even outperforming positional encoding in some cases.

Table 1: Results using ViT, MEGA and Swin backbones on the Tiny ImageNet (T-IN) and CIFAR-100 (C100) datasets. No hyperparameter tuning except for stochastic depth.

| Model | C100 | T-IN | Train Time | # Of Parameters |
|---|---|---|---|---|
| ViT | 73.81 | 57.07 | 1x | 2.71M (1x) |
| ViT w/ S4ND. | 72.60 | 56.10 | 2.22x | 2.72M (1.003x) |
| ViT w/ SSM-r. | **77.21** | **60.01** | 2.66x | 2.72M (1.003) |
| ViT w/ SSM-c. | 76.46 | 59.08 | 2.89x | 2.73M (1.01x) |
| Mega-ablate | 74.82 | 56.43 | 1.1x | 2.75M (1x) |
| Mega | 72.27 | 54.49 | 1.28x | 2.98M (1.08xx) |
| Mega w/ S4ND | 74.9 | 56.65 | 1.46x | 2.80M (1.02x) |
| Mega 2-D SSM-r | **76.02** | **57.95** | 2.03x | 2.80M (1.02x) |
| Mega 2-D SSM-c | 75.09 | 56.51 | 2.03x | 2.84M (1.03x) |
| Swin | 76.87 | 60.87 | 1x | 7.15M (1x) |
| Swin-reprod. | 77.98 | 61.29 | 1x | 7.15M (1x) |
| Swin w/ EMA | 77.01 | 60.13 | 1.39x | 7.52M (1.05x) |
| Swin w/ S4ND | 79.26 | 64.6 | 1.29x | 7.18M (1.004x) |
| Swin w/ SSM-r | **80.12** | **65.77** | 2.16x | 7.25M (1.01x) |
| Swin w/ SSM-c | 3.28 | 12.76 | 2.18x | 7.26M (1.02x) |

Table 2: ViT & Celeb-A resutls.

| Model | Top 1 | #Param |
|---|---|---|
| DeiT-T | 88.43 | 5.532M |
| DeiT-T w. SSM-r | 89.76 | 5.537M |
| DeiT-T w. SSM-c | **89.84** | 5.541M |
| DeiT-S | 89.66 | 21.681M |
| DeiT-S w. SSM-r | 90.24 | 21.688M |
| DeiT-S w. SSM-c | **90.38** | 21.691M |
| DeiT-B | 90.13 | 85.829M |
| DeiT-B w. SSM-r | 90.45 | 85.841M |
| DeiT-B w. SSM-c | **90.73** | 85.845M |
| Swin-T | 91.48 | 27.550M |
| Swin-T w. SSM-r | 91.68 | 27.556M |
| Swin-T w. SSM-c | **91.78** | 27.558M |

Table 3: ViT & ImageNet-100 runs with and without 2D-SSM

| Models | Baseline | w/ 2D-SSM |
|---|---|---|
| DeiT-T | 78.21 | **81.16** |
| DeiT-S | 82.27 | **84.82** |
| Swin-T | 81.06 | **82.29** |

**Controllable Unrestricted Spatial Context** A significant limitation of both CNNs and transformers is the need to choose an appropriate patch size, which can result in a loss of global or local context. In contrast, our layer implements a controllable global convolution, which benefits from a global context, and can control the effective receptive field and efficiently capture spatial local dependencies. Moreover, our layer is not confined to patch-like dependencies and can effectively capture diverse local and global features in horizontal, vertical, or diagonal directions (See 1, middle).

**Modeling Symmetries** Similar to other CNNs, our layer exhibits translation equivariance as the kernels slide across the input during computation. Additionally, as detailed in Appendix D, our layer's core extends over multiple directions, which allows it to accommodate rotations and reflections naturally.

**Parameter Efficiency** In contrast to CNNs, which parameterize filters of size $H \times W$ with at least $HW$ parameters, our layer has a fixed and small number of parameters (9 parameters per channel, $A_1, A_2, A_3, A_4, B_1, B_2, C_1, C_2, D \in \mathbb{R}$), however, those parameters can be expanded into unbounded two-dimensional kernels. Furthermore, we use parameter sharing for the SSM parameterization across channels, similarly to CNNs or state-space layers and donate $n_{ssm}$ as the number of non-shared channels.

## 5 EXPERIMENTS

We assess our 2-D SSM layer as an inductive bias within various ViT-based and CNN backbones. We demonstrate the universality of our method by incorporating it into various backbones, such as ViT, DeiT, Swin, ConvNeXt and report improved results over the baselines on ImageNet-1k, ImageNet-100, Celeb-A, Tiny-ImageNet and CIFAR-100, with negligible additional parameters and without hyperparameter tuning, except for stochastic depth. For a comprehensive overview of the experimental setup, please see Appendix F.

**Swin, DeiT and ViT** We adopted a straightforward approach to incorporate our 2-D SSM layer into the aforementioned backbone structures. Prior to the Transformer Block, we apply the 2-D SSM to the input signal $u \in R^{L_1 \times L_2 \times D}$, as illustrated in Fig. 1. We highlight that in the case of Swin Transformer, the 2-D SSM operates on the patch level and not the window level and therefore injects additional 2-D bias between windows.

We tested **Swin and ViT** over the small datasets Tiny-ImageNet and CIFAR-100, using the results reported by (Lee et al., 2021) as baseline. As shown in Tab. 1, with the ViT backbone, we improve $0.8\%$ on CIFAR-100 and $0.5\%$ on Tiny-ImageNet, and with the Swin backbone, we improve $3.25\%$ on CIFAR-100 and $4.25\%$ on Tiny ImageNet.

Table 4: Accuracy on the CIFAR-10 grayscale classification task, which is part of the Long Range Arena.

| Models | Image - LRA |
|---|---|
| Transformer | 42.94 |
| S4-v1 | 87.26 |
| S4-v2 | 88.65 |
| CCNN-1D | 88.90 |
| CCNN-2D | 91.12 |
| S4ND | 89.90 |
| Hyena | 91.20 |
| Mega Ablate | 81.00 |
| Mega | 90.44 |
| MEGA 2-D SSM | **91.31** |

Table 5: ImageNet-1K accuracy of MEGA variants.

| Model | Top 1 | Top 5 | # of Parameters |
|---|---|---|---|
| MEGA-ablate | 66.97 | 88.17 | 5.91M |
| EMA | 69.73 | 89.76 | 6.21M |
| 2D-SSM-R | **70.11** | **90.19** | 5.96M |

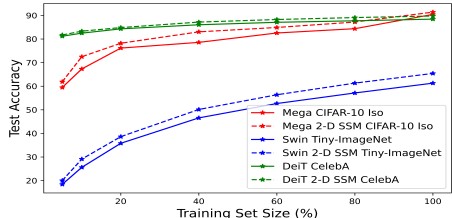

Figure 4: The effect of the training set size.

The **DeiT and Swin** backbones were tested on the large-scale Celeb-A dataset (Liu et al., 2015) and ImageNet-100. Celeb-A involves a 40-way multi-label attribute classification. We report aggregate accuracy across all 40 tasks. As can be seen in Tab. 2, the complex version outperforms the real in all experiments and achieves $1.41\%$, $0.72\%$, $0.6\%$, and $0.3\%$ improvements over the baseline for DeiT sizes Tiny, Small, and Base, and Swin-Tiny respectively. The ImageNet-100 benchmark is a known subset of ImageNet-1K. The results of these experiments are presented in Tab. 3. The hyperparameters for each model are taken from the corresponding paper and git repository. For DeiT, our 2-D SSM layer resulted in a performance enhancement of 2.95% (from 78.21% to 81.16%) on DeiT-T and 2.55% (from 82.27% to 84.82%) on DeiT-S. On the Swin-T, the performance improved by 1.23%, elevating the accuracy from 81.06% to 82.29%.

**Mega**  In Mega, we replace the EMA mechanism with 2-D SSM, which means that we only perform our layer on $Q, K$. We compare original Mega (with EMA) vs Mega-ablate (without EMA) and Mega 2-D SSM. We examined our model on CIFAR-10 Grayscale in an isotropic manner (without decreasing the image size along the architecture, and without patches), which is part of the Long Range Arena benchmark (Tay et al., 2020). As shown in Tab. 4, we improved the result by almost 1% over MEGA, obtaining state-of-the-art results, including very recent approaches such as Hyena (Poli et al., 2023) and CCNN-2D (Knigge et al., 2023). We also conduct an experiment on the ImageNet-1K dataset (Deng et al., 2009), and as shown in Tab. 5, we improve over MEGA's ViT-T results by $\sim 0.4\%$ in both Top 1 accuracy and Top 5 accuracy. Finally, we check other small datasets (Tab. 1) and find superior results for combining Mega with 2-D SSM over baseline or other methods.

**ConvNeXt**  In ConvNeXt, we replace the $7 \times 7$ Conv2D filters with 2-D SSM layers. We evaluate our ConvNeXt variants on four datasets, including high-resolution (Imagenet-100 and Celeb-A) and low-resolution (Tiny-Imagenet and CIFAR-100) datasets, across three model sizes: ConvNeXt-Tiny, ConvNeXt-Micro, and ConvNeXt-Nano. These backbones were introduced in previous work, the differences between the backbones, and more details about the experimental setup are provided in Appendix G. As highlighted in Tab. 6 and Tab. 7, both our complex and real variants consistently outperform the original ConvNeXt across all datasets and model sizes, without changing the parameter amount significantly. It is important to note that we do not perform hyper-parameter tuning, instead, we use the exact same hyper-parameters optimized for the CNN baseline, yet our results achieve SOTA performance on ImageNet-100 and CelebA datasets.

| Model | IM-100 | Celeb | # Param |
|---|---|---|---|
| T-Conv2D | 89.72 | 91.70 | 27.90M |
| T-2D SSM.R | **90.81** | 91.81 | 27.68M |
| T-2D SSM.C | 90.47 | **91.96** | 27.79M |
| M-Conv2D | 88.51 | 91.00 | 9.27M |
| M-S4ND | N.A | 91.30 | 9.60M |
| M-2D SSM.R | **89.81** | 91.61 | 9.23M |
| M-2D SSM.C | 89.16 | **91.75** | 9.33M |

Table 6: Micro (M) and Tiny (T) ConvNeXt results on Imagenet-100 and Celeb-A.

| Model | C100 | T-IM | # Param |
|---|---|---|---|
| N-Conv2D | 79.10 | 67.09 | 6.48M |
| N-2D SSM.R | **80.16** | **69.10** | 6.51M |
| N-2D SSM.C | 79.92 | 67.46 | 6.63M |

Table 7: Results of ConvNeXt-Nano on Tiny-Imagenet (T-IM) and CIFAR-100 (C100).

Table 8: Ablations. For each model and dataset, we examine the effect of using original positional encoding and complex (C) vs. real (R) SSM. The column $\delta$ represents the average difference for models with and without PE. As can be seen, our models are much more resistant to PE removal.

| Dataset: | Tiny-INet (Swin) | | CIFAR100 (ViT) | | CelebA (DeiT-T) | | CIFAR10 (Mega-ISO) | | Avg. |
|---|---|---|---|---|---|---|---|---|---|
| Model | with PE | w/o PE | with PE | w/o PE | with PE | w/o PE | with PE | w/o PE | $\delta$ |
| Baseline | 61.29 | 58.97 | 73.26 | 64.09 | 88.43 | 87.99 | 90.44 | 75.21 | -6.79 |
| +Ours (R) | **65.77** | 65.44 | 74.07 | **74.89** | 89.76 | 89.63 | **91.31** | 90.68 | -0.07 |
| +Ours (C) | 3.28 | 2.16 | 73.91 | 74.67 | **89.84** | 89.83 | 90.46 | 90.79 | **-0.01** |

**Comparisons against S4ND**  S4ND is the only N-Dimensional SSM-based layer known to us. Nguyen et al. (2022) reported results on Celeb-A with ConvNeXt-Micro. As shown in Tab. 6, our real and complex variants show improvements over S4ND by 0.31% and 0.45%, respectively, both with a smaller number of parameters. We further compare our results against it by substituting SSM in ViT, Mega, and Swin. We conducted experiments on CIFAR100 and Tiny Imagenet. As indicated in Tab. 1, S4ND performs very poorly when integrated into the original ViT backbone on CIFAR-100 (lower than the baseline by 1.21%) and sub-optimally when integrated into Swin and MEGA (achieves 1% or more lower accuracy on CIFAR-100 and Tiny-ImageNet for both backbones).

**Sample Complexity**  We examine the behavior of our model with different backbones on different datasets over the baseline. As can be seen in Fig. 5, 2-D SSM maintains improved results over the baseline for all backbones, which shows the data-efficient quality of our model.

**Removing the positional encoding (PE)**  We compare the empirical results obtained with real vs. complex kernels, with and without PE in Tab. 8. Evidently, complex-SSM can be superior or inferior to real-based SSM, depending on the scenario. Additionally, we find that complex-SSM has a tendency to exhibit instability during training, which can result in poor performance. Stabilizing these models is an important direction for future research.

Running ViT backbones without PE decreases performance dramatically. In contrast, when our 2D-SSM layer is inserted into these backbones, they benefit from PE, and even without PE they outperform the original backbones with PE. These findings support an innovative approach to introducing positional bias in ViT: rather than encoding positional information directly into the representation, it is integratede into the computation by incorporating positional-dependent operators.

# 6  LIMITATIONS

Despite the promising results presented, the current implementation of our proposed layer has relatively slow training times. One possible approach to mitigating this challenge is to use multidimensional parallel scanners, which could potentially reduce the training time of our layer. The main idea is extending the work of S5 (Smith et al., 2022), which leverages 1-D parallel scanners to apply SSM on 1-D sequences to multi-dimensional parallel scanners and multi-dimensional sequences.

# 7  CONCLUSIONS

We present a novel spatial SSM-based layer that is more general than existing ones, encodes positional information by design, and is able to model spatial relations more expressively than other SSMs, including S4ND. When added to various backbones, it is able to improve classification results on the various benchmarks without optimizing any aspect or other parts of the architecture. In future work, we would like to study the behavior of the layer in the context of spatial vision tasks, such as video processing, image segmentation, phrase grounding, and image inpainting. In the last task, the recursive view of the layer could be applied directly to impute missing pixels efficiently.

## 8 ACKNOWLEDGMENTS

This work was supported by a grant from the Tel Aviv University Center for AI and Data Science (TAD), and the Blavatnik Family Foundation. This research was also supported by the Ministry of Innovation, Science & Technology ,Israel (1001576154) and the Michael J. Fox Foundation (MJFF-022407). The contribution of IZ is part of a Ph.D. thesis research conducted at Tel Aviv University.

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

# A  COMPUTING THE KERNEL

We discuss $x_{i,j}^h, k_{i,j}^h$. The same calculations hold for $x_{i,j}^v, k_{i,j}^v$.

$k_{i,j}^h$ can be written as:

$$\forall i, j : k_{i,j}^h = \sum_{z}^{2*L_{max}} c_z A_1^{z_1} A_2^{z_2} A_3^{z_3} A_4^{z_4} B_{z_5} \tag{15}$$

For brevity and since it is not material for the method, we limit our exposition of the different power combinations of the system matrices $A_1, A_2, A_3, A_4$ and the input matrices $B_1, B_2$. As noted above, for each $k_{i,j}^h$ there are at most $2L_{max}$ elements.

**Pre-Processing**  The problem of finding $c_z$ for each element in the summation is a generalization of Pascal's triangle. In order to calculate the kernel, we calculate all the coefficients and the powers of $A_1...A_4$ up to the size of $L_1, L_2$ and cache them before the training process.

During training, we employ a matrix multiplication process to compute $k_{i,j}^h$ with the learned parameters $A_1, ..., A_4, B_1, B_2$.

Thus, for each cell there are at most $2L_{max}$ elements, and for each element, we save a constant number of $\chi$ values (the values of $z$ and $c_z$). As a result, the size of the cached matrix is bounded by $\mathcal{O}(L_{tot}L_{max})$.

It should be noted that currently in our method we cache the coefficients as One Hot Encoding and not the coefficient itself, and thus in our specific implementation we need to multiply the time complexity and memory complexity by $L_{max}$.

# B  TIME AND MEMORY COMPLEXITY OF OUR METHOD

To understand the complexity of our method, we will outline each step of the computation, starting from calculating the cache, creating the kernel during training, and computing the output $Y$ afterward.

As before, for brevity, we will refer only to the $x_{i,j}^h, k_{i,j}^h$ matrices caching, but the same holds for the vertical matrices. For simplicity, we assume $H = 1$ (number of channels).

**Caching**  As noted in Section 3.3, for each cell $k_{i,j}^h$ in the horizontal kernel there are at most $2L_{max}$ elements. Also as noted in Section 3.3, $z_1, z_2, z_3, z_4 <= 2L_{max}$. Thus, for each element in Eq. 15 we save $z_1, z_2, z_3, z_4, z_5$ and $\alpha_z$ values. In total, for each cell, we save $\chi_1 L_{max}$ values, where $\chi_1$ is a small constant. We have $L_{tot}$ cells and thus the total coefficient cached tensor for calculating $K_h$ is sized

$$\chi_1 * L_{max}L_{tot} \tag{16}$$

From now on we will denote the tensors of horizontal coefficients as $CACHE \in \mathbb{R}^{\chi_1 \times L_{tot} \times L_{max}}$. Notice that there is a $CACHE$ tensor for each parameter, meaning $CACHE_{A_1}^h, CACHE_{A_2}^h, CACHE_{A_3}^h, CACHE_{A_4}^h, CACHE_B^h$.

**Creating the Kernel**  For brevity, we use real-valued diagonal $A_i \in [0, 1]^N$ (after the sigmoid). First, we calculate the Vandermonde Matrix for each $A_1, A_2, A_3, A_4$ eigenvalues up to the highest power that exists in the kernel, which is $2L_{max}$, and denote $VAN_i = Vandermonde(A_i) \in \mathbb{R}^{2L_{max} \times N}$.

Again, we have $L_{tot}$ cells in the horizontal kernel, each cell having $2 * L_{max}$ elements. We take $CACHE_{A_i}$ which holds for each element its $A_i$ power, $z_i$ and creates a matrix that holds for each element its corresponding $A_i^{z_i}$ value.

$$O_{A_i}^h = VAN_i[CACHE_{A_i}^h] \in \mathbb{R}^{L_{tot} \times 2L_{max} \times N} \tag{17}$$

Now we multiply the matrices element-wise to obtain the final value of each element:

$$O_{pre-addition}^h = O_{A_1}^h \odot O_{A_2}^h \odot O_{A_3}^h \odot O_{A_4}^h \odot O_B^h \odot O_\alpha^h \in \mathbb{R}^{L_{tot} \times L_{2L_{max}} \times N} \tag{18}$$

where $\odot$ denotes element-wise multiplication. Now for each cell, we sum all the elements in the summation $k_{i,j}$, meaning summing over the second dimension:

$$O_{post-addition}^h = sum(O_{pre-addition}^h, d = 1) \in \mathbb{R}^{L_{tot} \times N} \tag{19}$$

Again, all the above steps are employed for the vertical axis as well, thus we are finally able to compute the kernel by using $C_1, C_2 \in \mathbb{R}^{N \times 1}$:

$$K = O^h_{post-addition}C_1 + O^v_{post-addition}C_2 \in \mathbb{R}^{L_1 \times L_2} \tag{20}$$

Remembering that we actually used $n_{ssm}$ channels, the kernel size is $K \in \mathbb{R}^{L_1 \times L_2 \times n_{ssm}}$. It should be noted here that the calculation of the kernel is not dependent on the batch size $B$.

**Forward Pass**    Let B denote the batch size. We would like to convert input signal $U \in \mathbb{R}^{B \times L_1 \times L_2 \times H}$ to output signal $Y \in \mathbb{R}^{B \times L_1 \times L_2 \times H}$. After calculating the kernel, we convert $U, K$ to the frequency domain through FFT:

$$U_f = FFT(U), K_f = FFT(K) \tag{21}$$

$$Y = IFFT(U_f \odot K_f) \tag{22}$$

This whole process costs us $O(BHL_{tot} \log(L_{tot}))$

Thus, our total forward pass time complexity is:

$$\mathcal{O}(\chi L_{TOT} L_{max} n_{ssm} N + BHL_{TOT} \log L_{TOT}) \tag{23}$$

**Implementation detail**    We implemented the caching and matrix multiplication process with One Hot Encoding Vector of the powers and not by using floats representing the powers themselves. Thus, the size of each $COEFF^h_i$ in our implementation is multiplied by $L_{max}$, as is the time complexity of Eq. 17 and 18.

## C    EXPRESSIVENESS

**Theorem C.1.** *One channel of* 2-D SSM *can express full-rank kernels*

*Proof.* We start by restricting the system, output and input matrices:

$$A_1 = A_2 = A_3 = 1, \quad A_4 = 0, \quad C_1 = 1, C_2 = 0, \quad B_1 = 1, B_2 = 0 \tag{24}$$

For simplicity we assume that the values of the initial states are 0:

$$\forall i : i = -1 \rightarrow x_{i,j} = 0, \quad \forall j : j = -1 \rightarrow x_{i,j} = 0 \tag{25}$$

It suffices to show that (i) $K$ is a triangular matrix, and (ii) the diagonal of $K$ contains non-zero elements. First, by plugging 24,25 into the recurrent rule 2, it can be simplified:

$$y_{i,j} = x^h{}_{i,j}, \quad x^h{}_{i,j} = x^h{}_{i,j-1} + x^v{}_{i,j-1} + u_{i,j}, \quad x^v{}_{i,j} = x^h{}_{i-1,j} \tag{26}$$

Given this simplification 26, both (i) and (ii) can be easily proven by induction on the diagonals of $K$.

To provide more insight into the proof, Eq. 27 illustrates the values of $K$.

$$\begin{bmatrix} 1 & 1 & 1 & 1 & 1 \\ 0 & 1 & 2 & 3 & 4 \\ \vdots & \ddots & 1 & 3 & 6 \\ \vdots & \ddots & \ddots & 1 & 4 \\ 0 & \cdots & \cdots & 0 & 1 \end{bmatrix} \tag{27}$$

And in general, since its clear from 26 that

$$y_{i,j} = x^h{}_{i,j} = y_{i,j-1} + y_{i-1,j-1} + u_{i,j}, \quad \forall j \rightarrow k_{0,j} = 1 \tag{28}$$

It easy to understand that the upper triangular of $K$ can obtained from a rotation of Pascal's triangle.

$\square$

## D  MODEL EXTENSION

### D.1  BIDIRECTIONAL SSM

Using the State Space model, when calculating $x^h_{i,j}$ one only considers $x_{\hat{i},\hat{j}}$ where $\hat{i} \leq i, \hat{j} \leq j$. To benefit from bidirectionality, we employ a version where we transpose the kernel in two or four directions (shown in the ablation experiments) and then sum the results. A similar mechanism for the 1-D case is used in S4, MEGA, and elsewhere.

### D.2  MULTI-AXIS MULTIDIMENSIONAL SSM

To enrich the dependencies that can be captured by our kernels, we use a weighted linear combination of kernels. Under the assumption that the system matrices are diagonal, the $N$ coordinates of the states are not dependent on each other, and can thus be calculated independently. Specifically, Eq. 9 can be re-written separately per coordinate : $\forall g \in [N]$

$$x^h{}_{i,j}[g] = \sum_{0 \leq \hat{i} \leq i} \sum_{0 \leq \hat{j} \leq j} k^h{}_{\bar{i},\hat{j}}[g] u_{\hat{i},\hat{j}} \tag{29}$$

Therefore, increasing $N$ will increase the number of kernels that make up $K$. By using this structure, the kernel can capture a variety of dependencies, where each coordinate focuses on a different type of dependency. Furthermore, this extension adds relatively negligible runtime when working with large batches, since the batch dimension does not affect the complexity when the kernel is computed. Therefore, increasing $N$ will increase the number of kernels that compose $K$.

## E  JUSTIFY DESIGN CHOICES

### E.1  OUR COMPLEX 2D-SSM

As explained in Sec. 3.3, our models employ real rather than complex SSMs. For reproducibility, here we provide a detailed description of our complex SSM variant: The complex variant of our 2-D SSM model, still assumes $\forall t, u_{i,j} \in \mathbb{R}^1$ , $\forall t, y_{i,j} \in \mathbb{R}^1$ and employs:

$$A_1, A_2, A_3, A_4 \in \mathbb{C}^{NxN}, B_1, B_2 \in \mathbb{C}^{Nx1}, C_1, C_2 \in \mathbb{C}^{1xN} \tag{30}$$

(diagonal matrices as above), and therefore

$$x^h_{i,j} \in \mathbb{C}^N, x^v_{i,j} \in \mathbb{C}^N \tag{31}$$

The output remains a real number $y_{i,j} \in \mathbb{R}^1$, and thus the real part of Eq. 2 is used, namely $y^{out}_{i,j} = \mathrm{Re}(y_{i,j})$

For complex SSM, we save $\hat{A}_{i_{angle}}, \hat{A}_{i_{radius}} \in \mathbb{R}^{N \times N}$, and we calculate $A_i \in \mathbb{C}^{N \times N}$ in the following manner:

$$A_{i_{angle}} = 2\pi sigmoid(\hat{A}_{i_{angle}}), \quad A_{i_{radius}} = sigmoid(\hat{A}_{i_{radius}}) \tag{32}$$

$$A_i = A_{i_{radius}} * (cos(A_{i_{angle}}) + i * sin(A_{i_{angle}})) \in \mathbb{C}^{NxN} \tag{33}$$

The same goes for $B_1, B_2$. As for $C_1, C_2$, we perform the same operation without limiting the radius size (not applying a sigmoid to $\hat{C}_{i_{radius}}$).

### E.2  NO WEIGHT DECAY ON THE SSM CORE

While vision transformers and MEGA regularize the model via weight decay, SSM-based layers typically do not apply this (Gupta, 2022; Gu et al., 2021a), since it drastically lowers the models' ability to learn, especially in the context of long-range dependencies. In general, higher values of the $A_i, B, C$ parameters do not seem to correspond with overfitting. Therefore, our method does not employ weight decay on those parameters.

## F    EXPERIMENTAL SETUP

We use PyTorch for all experiments. As a deliberate decision we choose to not perform hyper-parameter tuning of the backbone and training procedure, apart from stochastic depth. All experiment results were averaged over seeds = $[0, 1, 2]$. For all datasets and backbones, we set $n_{ssm} = 8, N = 16$ for all SSM-Based variants (SSM-2D real & complex and S4ND).

**Cifar-100 and Tiny imagenet**   For both datasets, we use as a baseline the experiments performed by (Lee et al., 2021). This means we follow DeiT's (Touvron et al., 2021) application of a long list of data augmentation and regularization methods, including Cutmix (Yun et al., 2019), Mixup (Zhang et al., 2017), stochastic depth (Huang et al., 2016), repeated augmentation (Hoffer et al., 2020), Rand-Augment (Cubuk et al., 2020), and random erasing (Zhong et al., 2020). AdamW was used as the optimizer. Weight decay was set to 0.05 (apart from SSM layer where it was set to 0), batch size to 128, and warm-up to 10. All models were trained for 100 epochs, and cosine learning rate decay was used. The initial learning rate was set to 0.003. In certain scenarios, we noticed that our models converge faster compared to the baseline approach. We discovered that a slight modification, specifically doubling the stochastic depth, proved to be instrumental in maximizing the model's performance.

When comparing S4ND, we used the same parameter scheme being used in 2-D SSM to perform a valid comparison, by making $C \in \mathbb{C}^{n_{ssm}, N}$ instead of $C \in \mathbb{C}^{H, N}$ as in the original paper.

**CelebA**   For Celeb-A, the original image size is 178x218, it is resized to 224x224 to match DeiT (Touvron et al., 2021) backbone and patch size. The dataset includes a 40-way multi-label attribute classification. We are reporting an average accuracy of all 40 tasks. We use the same data augmentation, and hyperparameters as DeiT, and train the models for 20 epochs, similar to the training procedure of S4ND (Nguyen et al., 2022) on this datasets.

**Imagenet and CIFAR-10 Grayscale**   We use the exact same training procedure including hyper-parameters, data augmentation and training environment as used in the git repository of the baseline (Ma et al., 2022) for those datasets.

## G    CONVNEXT VARIANTS

The ConvNeXt model marks a significant advancement in CNN architectures, building upon the foundation of classic CNNs such as ResNets and inspired from the progress in ViT. It incorporates three main areas of enhancement: (i) micro design changes, including the adoption of improved activation functions and normalization methods. (ii) macro design adjustments, such as implementing depthwise convolutions with larger kernels and modern stage compute ratio, and (iii) a modern training regime that extends the training process, employs the AdamW optimizer, and incorporates advanced augmentation and regularization strategies like Stochastic Depth and Label Smoothing.

**ConvNeXt Variants**   In the original ConvNeXt paper Liu et al. (2022), five model sizes were proposed:

1. ConvNeXt-T (Tiny): $\dot{C} = (96, 192, 384, 768)$, $\dot{B} = (3, 3, 9, 3)$

2. ConvNeXt-S (Small): $\dot{C} = (96, 192, 384, 768)$, $\dot{B} = (3, 3, 27, 3)$

3. ConvNeXt-B (Base): $\dot{C} = (128, 256, 512, 1024)$, $\dot{B} = (3, 3, 27, 3)$

4. ConvNeXt-L (Large): $\dot{C} = (192, 384, 768, 1536)$, $\dot{B} = (3, 3, 27, 3)$

5. ConvNeXt-XL (Extra Large): $\dot{C} = (256, 512, 1024, 2048)$, $\dot{B} = (3, 3, 27, 3)$

where $\dot{C}$ represents the number of channels and $\dot{B}$ the number of blocks in each stage. Later, for low-resource environments and small dataset applications, two additional variants were proposed:

1. ConvNeXt-M (Micro): $\dot{C} = (64, 128, 256, 512)$, $\dot{B} = (3, 3, 3, 3)$

2. ConvNeXt-N (Nano): $\dot{C} = ([64, 128, 256, 512])$, $\dot{B} = (2, 2, 2, 2)$

The ConvNeXt-Micro variant was proposed in S4ND Nguyen et al. (2022) for the Celeb-A dataset, and the Nano variant, which introduced in this paper, is taken from this link, and was designed for the CIFAR-10 dataset.

**Experimental Setup for ConvNeXt**   For experiments with ConvNeXt-M we employ the exact hyperparameters used in Nguyen et al. (2022). In the case of ConvNeXt-T, we followed the hyperparameter specifications from the original ConvNeXt paper, with an exception for the number of training epochs in the Celeb-A experiments, which were adjusted to align with the baselines reported in the S4ND paper (20 epochs), ensuring a balanced comparison. Regarding ConvNeXt-N, our approach mirrored the training protocol found at con, which involves a 100-epoch training period. We applied this scheme to CIFAR-100 and Tiny-Imagenet datasets, diverging from the original experiment's use of CIFAR-10, but maintained the same hyperparameters.

## H   INFERENCE PERFORMANCE ANALYSIS

To provide a comprehensive analysis of the 2-D SSM layer, we empirically examined both its latency and memory footprint.

**Inference Profiling**   To assess the efficiency of integrating our model into a Vision Transformer (ViT) backbone, we analyzed the inference latency. Fig. 6 displays a pie chart profiling the latency within a single block, encompassing elements like the attention mechanism, 2-D SSM layer, normalization layers, residuals, and feed-forward networks (FFNs). Notably, our 2-D SSM layer accounts for an additional 20.5%.

We further compared the execution time and memory usage of our layer during inference with the Self-Attention mechanism. As illustrated in Fig. 7, we evaluated different 2-D input dimensions. Our layer introduces less overhead overall, and notably, as the 2-D sequence length extends (especially beyond 100), the performance of the SSM-2D layer appears to converge more favorably compared to self-attention.

Fig. 8 presents the memory consumption data in a logarithmic scale. Here, the same trend is observed: the attention mechanism's memory usage increases quadratically relative to the 2-D SSM layer as the sequence length (L) grows. For example, for image sized 130x130, the attention mechanism uses $2e4$MB while 2-D SSM only needs $5e2$MB.

## I   OTHER RELATED WORK

**Vision Transformer Backbones**   To demonstrate the versatility and efficacy of our 2-D SSM layer as a plug-and-play component compatible with various ViTs, we evaluate its performance when integrated into the following backbone architectures: **(i) ViT** The original ViT that employs self-attention on a 1-D sequence of patches; it used a learnable 1-D position encoding. **(ii) Swin** The Swin Transformer (Liu et al., 2021) refines ViT by incorporating hierarchical structure and local connections within windows. It employs a shifted windowing scheme and stage-wise processing to efficiently capture global context, which enhances its performance across various vision tasks. **(iii) Mega**   The Mega (Ma et al., 2022) model introduced a single-head gated attention mechanism enhanced with an exponential moving average, which imparts position-aware local dependencies to the attention mechanism. This approach addresses the limitations of the Transformer's attention mechanism, such as weak inductive bias. Mega demonstrates superior performance across a variety of tasks, including the Long Range Arena, neural machine translation, language modeling, and image and speech classification. **(iv) DeiT**   (Touvron et al., 2021) is an adaptation of ViT, incorporating a class-token designed for distillation purposes.

**The ConvNeXT Liu et al. (2022) Backbone**   is the state-of-the-art CNN architecture, which offers a viable alternative to the Transformer-based models while maintaining competitive accuracy and computational efficiency. In our experiments, we maintain the architecture and replace the convolutional layers with our 2-D SSM layer to evaluate our method in a cutting-edge setting. Additional details about the ConvNeXt model and the variants we use are provided in Appendix G

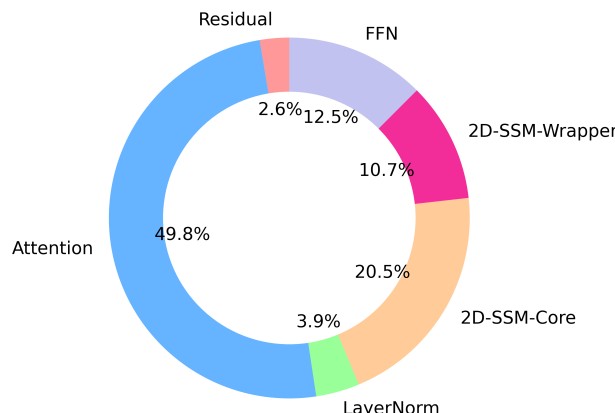

Figure 6: Pie chart depicting the inference time distribution for each component in a ViT backbone with an integrated 2-D SSM layer. '2-D-SSM-Core' represents operations directly associated with the layer, while '2D-SSM-Wrapper' encompasses surrounding operations like normalization and skip connections.

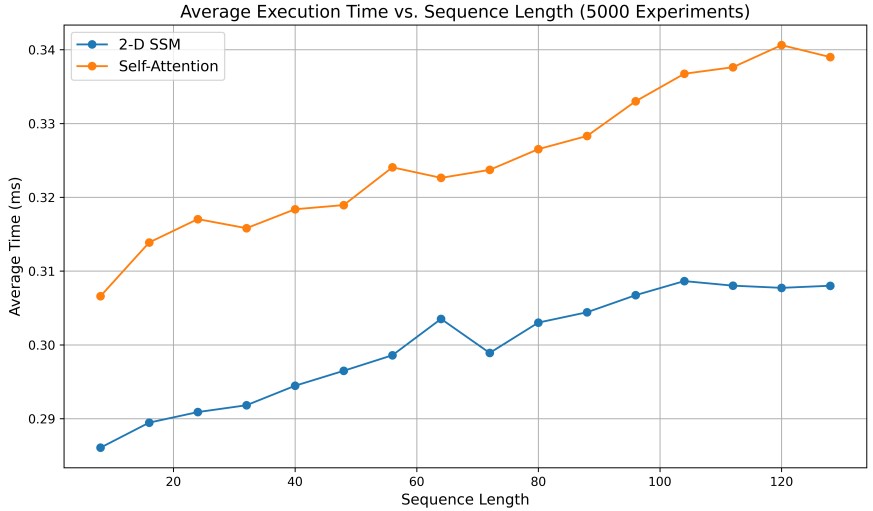

Figure 7: Comparison of Inference Time: Self-Attention mechanism vs. 2-D SSM, analyzed across various image sizes (L denotes one axis of the image, meaning image size is LxL).

**Adding positional bias into transformers**  By design, Vision Transformers are permutation invariant, and thus a lot of work was put into injecting bias into them. Besides the standard positional encoding, the following methods are proposed:

**Exponential Moving Average (EMA)**  The EMA is a common technique for smoothing time-series data and prioritizing recent data points. It is computed using $EMA_t = (1 - \alpha) \cdot EMA_{t-1} + \alpha \cdot u_t$, where $EMA_t$ is the current EMA value, $EMA_{t-1}$ is the previous value and $\alpha$ is the smoothing

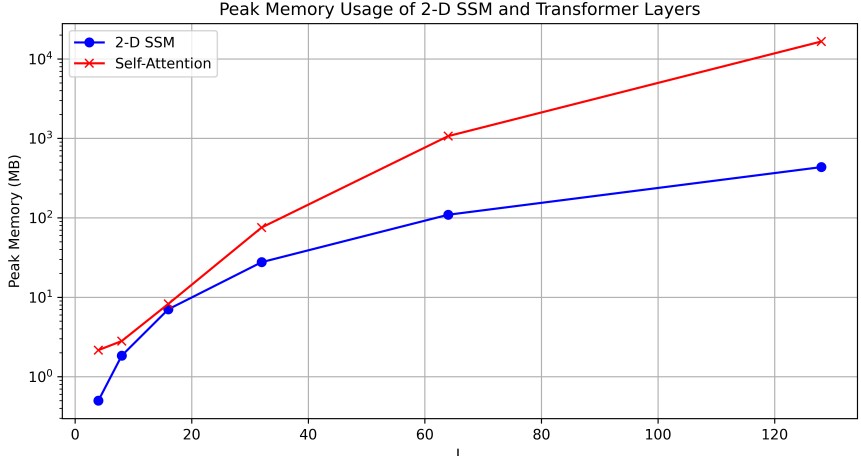

Figure 8: Comparison of memory consumption: Self-Attention mechanism vs. 2-D SSM, analyzed across various image sizes (L denotes one axis of the image, meaning image size is LxL), with memory usages (in mega bytes) presented in log-scale.

factor. It is being used in MEGA (Ma et al., 2022) to incorporate positional awareness bias into the attention.

**Other 2-D bias contributions**   By design, Vision Transformers are permutation invariant, and thus a lot of work was put into injecting 2-D bias into them. A particular research direction emphasizes the introduction of positional bias via various positional encoding methods. For instance, the Swin Transformer (Liu et al., 2021) employs a learnable bias term referred to as relative positional bias. In alternative avenues of research, efforts have been made to modify the attention window through diverse techniques, such as incorporating a two-dimensional local bias by cropping (Dong et al., 2021) the attention window or integrating convolutional neural networks (CNNs) with attention mechanisms (Dai et al., 2021), (Li et al., 2022a).

