# OpenReview forum: "A 2-Dimensional State Space Layer for Spatial Inductive Bias"
_ICLR.cc/2024/Conference — ICLR 2024 poster_

### Official Review · Reviewer_SZ7R · 2023-10-31

**Soundness:** 3 good
**Presentation:** 2 fair
**Contribution:** 3 good
**Rating:** 6
**Confidence:** 2

**Summary:**

The paper focuses on developing new method for injecting 2-D inductive bias into Vision Transformer for computer vision problems. To achieve this, the authors propose to leverage an expressive variation of the multidimensional State Space Model (SSM) with the proposed efficient parameterization, accelerated computation and suitable normalization scheme. The paper show that by incorporating the proposed layer at the beginning of each transformer block of ViT improves the performance of various ViT backbones, such as Mega, for various datasets for image classification. It is also shown that the method achieves effective results without positional encoding.

**Strengths:**

* The paper proposes a new method for encoding image-specific inductive bias for ViT in Computer vision problems.

* The proposed method is shown to be effective for Image Classification in various datasets with various ViT backbones with negligible amount of additional parameters and inference time.

* The method can achieve good performance without positional encoding.

**Weaknesses:**

* As mentioned by the authors, one major limitation of the method is its high training time cost. It can double the training time compared with the baseline, limiting its application to the training of large models on large benchmarks.

* The experiments limit to image classification problems and 2-D inductive bias can be very important for dense prediction. It would be better to also evaluate the proposed method for dense prediction problems like segmentation, depth estimation etc.

* The complex (C) variant of the proposed method may exhibit instability, obtaining very bad results, e.g. Table 1, Table 6. I would recommend the authors to include a detailed explanation of this situation and any potential ways of avoiding the instability.

**Questions:**

* I would suggest the authors to clearly explain each model in each table. For example, it would be much clear if the authors can explain what is 'ViT w/ MixFFN' and so on before describing the results. Also, are the results of the proposed method shown in Table 3 obtained by using SSM-r or SSM-c?

* It is great to see the analysis of the inference cost. I would recommend to also give comparisons of the inference time in Table 1. It is also suggested to include comparisons of the memory cost in Table 1 and the details about the platform for experiments.

---

> ### Author Response · Authors · 2023-11-19
>
> Thank you for the comprehensive review and the constructive feedback.
>
>
> > As mentioned by the authors, one major limitation of the method is its high training time cost. It can double the training time compared with the baseline, limiting its application to the training of large models on large benchmarks.
>
> We refer you to the main response, section 2 “Latency and Memory Analysis”.  In summary, first, we optimized our code, which resulted in a 30% decrease in inference time and a 25% reduction in training time. Second, for the ConvNeXt variants, our training time is only 1.26 times longer than the standard CUDA-dedicated and highly-optimized implementation of the Conv2D-based ConvNeXt (note that our code is not optimized for CUDA). Lastly, we wish to highlight that, from a theoretical perspective, our layer is quite efficient, as analyzed in Section 3.3 'Computation and Complexity,' and empirically in Appendix H.
>
> > The experiments limit to image classification problems and 2-D inductive bias can be very important for dense prediction. It would be better to also evaluate the proposed method for dense prediction problems like segmentation, depth estimation etc.
> In response to your suggestion regarding the evaluation of our method for dense prediction tasks such as segmentation and depth estimation, we acknowledge the significance of such assessments. However, it is important to note that our current resource constraints limit our ability to conduct extensive experiments in these areas, particularly due to the typical requirements for pretraining on large datasets like ImageNet-22\1K or the utilization of Self-Supervised Learning (SSL) techniques.
>
> Instead, we have chosen to demonstrate the effectiveness of our method by incorporating our layer into ConvNeXt (main response, section 1) and conducting additional experiments on multiple datasets. We believe that these experiments offer a different perspective on the efficacy of our approach and provide valuable insights.
>
>
> > The complex (C) variant of the proposed method may exhibit instability, obtaining very bad results, e.g. Table 1, Table 6. I would recommend the authors to include a detailed explanation of this situation and any potential ways of avoiding the instability.
>
> In all our experiments, we have maintained consistency by using the same hyperparameters that the backbone was originally trained with, without engaging in hyperparameter tuning. In the instances where instability was observed, such as in Table 1 and Table 6, we found that decreasing the learning rate to a smaller value (1e-4 instead of 1e-3) helped stabilize the training process.
>
> It is important to note that various architectures may indeed require different hyperparameters when applied to different datasets. However, we decided to retain the original hyperparameters in order to adhere to the framework's original setup without extensive hyperparameter adjustments.
>
>
> > I would suggest the authors to clearly explain each model in each table. For example, it would be much clear if the authors can explain what is 'ViT w/ MixFFN' and so on before describing the results. Also, are the results of the proposed method shown in Table 3 obtained by using SSM-r or SSM-c?
>
> Thank you for your feedback. We will remove the reference to the MixFFN experiment from our tables, as it was included erroneously. We will review and ensure all models in each table are clearly explained. Additionally, for any further clarification needed on specific experiments or tables, your guidance would be invaluable. Regarding Table 3, the table showcases the SSM-r version.

---

### Official Review · Reviewer_mKTe · 2023-11-01

**Soundness:** 3 good
**Presentation:** 2 fair
**Contribution:** 3 good
**Rating:** 6
**Confidence:** 3

**Summary:**

This paper aims to integrate spatial inductive biases into neural network architectures, such as Vision Transformers, through the application of a two-dimensional state space model (SSM). By incorporating certain assumptions, this method ensure that the computational complexity remains tractable. Experimental results demonstrate that the proposed method surpasses previous SSM baselines across a range of scenarios.

**Strengths:**

1. The introduction of a two-dimensional State Space Model (SSM) is an intuitive approach for incorporating spatial inductive biases within neural networks.
2. The suggested technique can be easily integrated into various neural network models.
3. Particularly in scenarios with small size of data, the proposed method demonstrates improved performance compared to baselines that are based on SSM.
4. I appreciate the comparison of the proposed method with S4ND in Section 4.1, as it effectively articulates the proposed method's strengths. Although a more generalized approach does not necessarily guarantee enhanced real-world expressiveness—occasionally it may even compromise the stability of the training process—the conducted experiments effectively demonstrate the proposed method's practicality.

**Weaknesses:**

1. One limitation of the proposed method, as highlighted in Section 6, is its computational complexity. The method approximately doubles the training time, imposing a considerable computational load. Furthermore, while the added complexity during inference may not substantially contribute to the overall computational demand, a detailed report of the actual inference times would be beneficial for a comprehensive understanding of the method's characteristics.
2. In the experiments, the proposed method is primarily benchmarked against SSM-based methods. However, various approaches employ convolution for embedding, for example in [1], and the use of convolutional layers or simple components for positional encoding is a common practice [2, 3, 4, 5]. It would be instructive to compare the proposed method with Vision Transformers that incorporate the methods to exploit spatial equivariance, which could serve as additional baselines. Furthermore, while it may not be a critical flaw, the implementation of the proposed method appears to be somewhat more complicated and less straightforward than simply employing convolutional layers.
3. I believe that a larger model and dataset size would more effectively leverage strong spatial equivariance [6]. This implies that the proposed method may not be effective in environments with substantial data and model scales. Considering that training on datasets like IN1K with 'Base' or larger models has become a norm in the era of Vision Transformers, the applicability of the proposed method in such standard real-world scenarios could be limited. Additionally, the focus of the experiments on smaller datasets and models further suggests potential constraints in its utility for larger-scale tasks.

.

[1] Xiao, Tete, et al. "Early convolutions help transformers see better." *Advances in neural information processing systems* 34 (2021): 30392-30400.

[2] Chu, Xiangxiang, et al. "Conditional positional encodings for vision transformers." arXiv preprint arXiv:2102.10882 (2021).

[3] Wu, Kan, et al. "Rethinking and improving relative position encoding for vision transformer." Proceedings of the IEEE/CVF International Conference on Computer Vision. 2021.

[4] Liu, Ze, et al. "Swin transformer v2: Scaling up capacity and resolution." Proceedings of the IEEE/CVF conference on computer vision and pattern recognition. 2022.

[5] Chu, Xiangxiang, et al. "Twins: Revisiting the design of spatial attention in vision transformers." Advances in Neural Information Processing Systems 34 (2021): 9355-9366.

[6] Gruver, Nate, et al. "The lie derivative for measuring learned equivariance." arXiv preprint arXiv:2210.02984 (2022).

---

**Post rebuttal**

Thank you for your efforts in conducting additional experiments. These further results have significantly improved the manuscript and mitigated many of its technical drawbacks. However, my concerns about the limited impact of this research persist. Therefore, I will retain my rating of 'Weak Accept'.

**Questions:**

1. Why is the `D` omitted in Equation 2? Does its inclusion empirically reduce the performance?
2. In Figure 4, it is observed that the 2-D SSM markedly enhances performance when 100% of the dataset is utilized, as opposed to 20% or less. This result is counterintuitive. Providing explanations for this phenomenon would enhance the comprehensiveness of the paper.

---

> ### Author Response · Authors · 2023-11-19
>
> Thank you for the comprehensive review and the constructive feedback.
>
> > In the experiments, the proposed method is primarily benchmarked against SSM-based methods. However, various approaches employ convolution for embedding, for example in [1], and the use of convolutional layers or simple components for positional encoding is a common practice [2, 3, 4, 5]. It would be instructive to compare the proposed method with Vision Transformers that incorporate the methods to exploit spatial equivariance, which could serve as additional baselines. Furthermore, while it may not be a critical flaw, the implementation of the proposed method appears to be somewhat more complicated and less straightforward than simply employing convolutional layers.
>
>
> Your suggestion to compare our method with Vision Transformers that utilize convolution for embedding and positional encoding is indeed valuable. We refer to the main response, part “2-D SSM as an alternative to standard Convolution layers (Experiments with ConvNeXt)”, where we outline the results of replacing the convolutional layer in ConvNext with our 2-D SSM layer. We believe this result showcases the effectiveness of our layer when compared to standard convolutional layers (which can be considered as representative cases of methods that exploit spatial equivariance).
>
>
> Addressing the perceived complexity of our proposed method, we understand it may initially seem complex relative to conventional convolutional layers. Yet, this complexity is offset by notable advantages. Our method employs a global 2-D kernel, ensuring a consistent parameter count regardless of kernel size. Alongside this, our parameterization technique results in kernels formed within a sub-vector space, leading to reduced noise interference. Our approach leads to enhanced performance over ViT & Convnext backbones, across all datasets, underscoring the efficacy of our more complex design.
> >I believe that a larger model and dataset size would more effectively leverage strong spatial equivariance [6]. This implies that the proposed method may not be effective in environments with substantial data and model scales. Considering that training on datasets like IN1K with 'Base' or larger models has become a norm in the era of Vision Transformers, the applicability of the proposed method in such standard real-world scenarios could be limited. Additionally, the focus of the experiments on smaller datasets and models further suggests potential constraints in its utility for larger-scale tasks.
>
>
>  Please refer to 'Nature of our Benchmarks' in the main response for more details. We acknowledge the value of such experiments; however, we believe that our extensive evaluation across six datasets, with several backbones (ConvNeXt, ViT, Swin, Mega, DeiT), and a variety of model sizes—from DeiT/Mega-Base (85M params) and Swin-T (27M params) to ConvNeXt-Micro (9M params)—provides a sufficient assessment to justify the empirical contribution of our 2-D SSM layer.
>
> >Why is the D omitted in Equation 2? Does its inclusion empirically reduce the performance?
>
>
> This was done solely for brevity. We interpret 'D' as a parameter-based skip-connection in our model. Noting that key papers, like [2], introduce the State-Space Layer with 'D' and later omit it, we agree it could be clearer to present the full equation initially. We are open to making this change for enhanced clarity in our paper.
> > In Figure 4, it is observed that the 2-D SSM markedly enhances performance when 100% of the dataset is utilized, as opposed to 20% or less. This result is counterintuitive. Providing explanations for this phenomenon would enhance the comprehensiveness of the paper.
>
>
> It seems that there may be a misinterpretation. The dotted line in the test accuracy graph represents the 2-D SSM and is consistently above the straight line, which indicates the baseline, at all data points. This shows that the 2-D SSM performs better across all dataset utilization levels. Furthermore, we use the same hyperparameters for all experiments. Those hyperparameters were optimized for the baseline (attention without our layer) for the full-dataset regime, which could explain such phenomena.
> [1]: SegFormer: Simple and Efficient Design for Semantic Segmentation with Transformers. Enze Xie, Wenhai Wang, Zhiding Yu, Anima Anandkumar, Jose M. Alvarez, Ping Luo
> [2]: Efficiently Modeling Long Sequences with Structured State Spaces. Albert Gu, Karan Goel, Christopher Ré

---

> ### Author Response · Authors · 2023-11-23
>
> > In the experiments, the proposed method is primarily benchmarked against SSM-based methods. However, various approaches employ convolution for embedding, for example in [1], and the use of convolutional layers or simple components for positional encoding is a common practice [2, 3, 4, 5]. It would be instructive to compare the proposed method with Vision Transformers that incorporate the methods to exploit spatial equivariance, which could serve as additional baselines. Furthermore, while it may not be a critical flaw, the implementation of the proposed method appears to be somewhat more complicated and less straightforward than simply employing convolutional layers.
>
> In our earlier response, we added a comparison to a layer with a spatial inductive bias by comparing it with Conv2D within the state-of-the-art ConvNeXt. We would like to introduce another set of experiments using a ViT variant called MixFFN from SegFormer [1]. MixFFN omits positional encoding and integrates a 3x3 depth-convolution within the MLP, specifically before the hidden layer. In the table below, our modified approach—where the 2-D SSM is placed within the MLP in a manner akin to MixFFN, rather than preceding the attention layer and excluding the positional encoding—demonstrates superior performance over MixFFN. We tested it on CIFAR-100 and Tiny-ImageNet.
>
>
>
> | Dataset            | Mix-FFN | w/2-D SSM | Delta | #Params Ratio |
> |--------------------|---------|-----------|-------|---------------|
> | Tiny-ImageNet | 58.94   | 60.01     | 1.07  | 1.0035x       |
> | CIFAR-100   | 76.32   | 77.21     | 0.89  | 1.0035x       |
>
> [1]: SegFormer: Simple and Efficient Design for Semantic Segmentation with Transformers. Enze Xie, Wenhai Wang, Zhiding Yu, Anima Anandkumar, Jose M. Alvarez, Ping Luo

---

### Official Review · Reviewer_ZC1E · 2023-11-01

**Soundness:** 4 excellent
**Presentation:** 3 good
**Contribution:** 3 good
**Rating:** 6
**Confidence:** 3

**Summary:**

The submission introduces a novel layer based on a variation of the multidimensional State Space Model (SSM), aimed at enhancing 2-D inductive bias in computer vision models. The 2D-SSM layer is designed to be integrated into Vision Transformers (ViT), contributing to improved model performance across various ViT backbones and datasets, without adding substantial parameters. The authors underscore the layer’s ability to bring about strong 2-D inductive bias, highlighting its performance even in the absence of positional encoding, and showcasing its robustness through ablation studies and visualizations.

**Strengths:**

The central innovation of this work is the 2D-SSM layer, which is grounded in Roesser’s model for multidimensional state space, and benefits from efficient parameterization, accelerated computation, and suitable normalization. The layer introduces a strong inductive bias toward 2-D neighborhood and locality, captures unrestricted controllable context, and is highly parameter-efficient, being able to express kernels of any length via just eight scalars. Through empirical evaluation, the authors demonstrate that their layer acts as a general-purpose booster for vision transformers, surpassing standard methods like positional encoding in effectively integrating positional bias, all the while maintaining efficiency in terms of parameters and computation at inference.

The work is well-grounded in control theory, with the authors providing theoretical analysis to show that their layer generalizes S4ND and exhibits greater expressiveness. The submission includes supplementary code, facilitating reproducibility and practical application of the proposed method. Overall, this work presents a significant contribution to the field of computer vision, introducing a novel layer that addresses key challenges in 2-D inductive bias and demonstrates notable performance enhancements for Vision Transformers.

**Weaknesses:**

1. **Extended Training Time**: The submission raises concerns about the extensive training time required for the proposed method. There is ambiguity regarding whether other methods, if given a comparable increase in computational resources, would yield indistinguishable results. The current results lack persuasive power as they do not strictly control factors, making it difficult to definitively attribute performance gains to the proposed method.

2. **Simplicity of Tasks**: The tasks used to evaluate the method are considered too simple, raising questions about whether the inductive bias introduced is specifically beneficial for such tasks. A more critical evaluation would involve assessing the performance of the baseline methods after large-scale pre-training to assess if the learned inductive bias during pretraining can obtain better performance in downstream tasks after finetuning (etc., prompt tuning). If such an approach yields better results, the practical significance of the proposed 2D-SSM becomes unclear.

3. **Limited to Transformer-based Architectures**: The method primarily targets transformer structures that lack explicit inductive bias design. It is uncertain how well the method would perform on convolutional networks (ConvNets) and how it compares to similar inductive bias methods, such as ConvNeXt. A comprehensive evaluation across different network architectures and inductive bias strategies is needed to fully understand the method's applicability and effectiveness.

**Questions:**

Please refer to the weakness

---

> ### Author Response · Authors · 2023-11-19
> **Official Comment by Authors**
>
> Thank you for the comprehensive review and the constructive feedback.
> > “Weaknesses: Extended Training Time”
>
> Please refer to  2. 'Latency and Memory Analysis” in the shared response for more details. In summary, first, we optimized our code, which resulted in a 30% decrease in inference time and a 25% reduction in training time. Second, for the ConvNeXt variants, our training time is only 1.26 times longer than the standard CUDA-dedicated and highly-optimized implementation of the Conv2D-based ConvNeXt (note that our code is not optimized for CUDA). Lastly, we wish to highlight that, from a theoretical perspective, our layer is quite efficient, as analyzed in Section 3.3 'Computation and Complexity,' and empirically in Appendix H.
>
>
> > "Weaknesses: Simplicity of Tasks: .. A more critical evaluation would involve assessing the performance of the baseline methods after large-scale pre-training ... can obtain better performance in downstream tasks after finetuning (etc., prompt tuning) ..”
>
> Such experiments rely on large pre-trained models (trained on ImageNet-22\1K or through SSL schemes), and therefore they are very costly and are out of our budget’s reach. Please refer to 'Nature of our Benchmarks' in the main response for more details. We acknowledge the value of such experiments, however, we believe that our extensive evaluation across six datasets, with several backbones (ConvNeXt, ViT, Swin, Mega, DeiT), and a variety of model sizes—from DeiT/Mega-Base (85M params) and Swin-T (27M params) to ConvNeXt-Micro (9M params)—provides a sufficient assessment to justify the empirical contribution of our 2-D SSM layer.
>
> > “Limited to Transformer-based Architectures: .. It is uncertain how well the method would perform on convolutional networks (ConvNets) and how it compares to similar inductive bias methods, such as ConvNeXt. A comprehensive evaluation across different network architectures and inductive bias strategies is needed to fully understand ..”
>
> Following the review, we have added results using the ConvNeXt backbone across four datasets and three model sizes. Please refer to the section '2-D SSM as an Alternative to Standard Convolution Layers (Experiments with ConvNeXt)' in the main response, and to Tables 6 and 7 in the revised manuscript for detailed information. As the results show, our 2-D SSM layer outperforms the Conv2D layers in these backbones for all tested experiments. We consider these results very promising and extend our gratitude to the reviewer for this constructive suggestion.

---

### Official Review · Reviewer_CN5B · 2023-11-05

**Soundness:** 3 good
**Presentation:** 3 good
**Contribution:** 3 good
**Rating:** 6
**Confidence:** 2

**Summary:**

The paper found that a 2D recurrent state space model (SSM) can be computed as convolution and proposed a SSM based layer which can be seamlessly plug into ViT.  Experiments show this new layer can improve ViT classification accuracy slightly.

**Strengths:**

1/ A new SSM based layer which has sound theoretical justification (I did not check the math carefully), and can be calculated as Convolutions.
2/ This SSM based layer can be easily plugged into ViT and good results are achieved.

**Weaknesses:**

Although it seems the new layer is based on sound theoretical justification, and experiment results show that it indeed works, the improvement is tiny and it's hard to see the real benefits of the proposed idea. Actually this tiny improvement may disappear when hyper parameters vary a little bit. Thus it is really a stretch to claim the benefits of the new layer. More experiments will be needed to justify.

**Questions:**

Besides image level classification, can you run experiments on pixel level tasks such as semantic segmentation or instance segmentation by plug in this new layer into, e.g., Mask2Former or Pyramid Vision Transformer?

---

> ### Author Response · Authors · 2023-11-19
> **Official Comment by Authors**
>
> Thank you for the comprehensive review and the constructive feedback.
>
>
> > “..  the improvement is tiny and it's hard to see the real benefits of the proposed idea. Actually this tiny improvement may disappear when hyper parameters vary a little bit”
>
> We believe that demonstrating performance improvements across six datasets (ImageNet, ImageNet-100, Celeb-A, Tiny ImageNet, CIFAR-100, and CIFAR-10), multiple backbones (ViT, Swin, Mega, DeiT, and isotropic designs), and various model sizes (such as DeiT-T, DeiT-S, DeiT-B) provides compelling evidence of our method's consistent value. Furthermore, our improvements achieved SOTA results on some of the benchmarks (Imagenet-100 and Celeb-A) and remained consistent during our additional experiments (see ablations in Figure 4 and Table 8). Moreover, we wish to reemphasize that for most benchmarks, we did not perform any hyper-parameter tuning, instead, we used configurations optimized for the baseline method. The only exception was in specific cases where we increased the stochastic depth, which we consider as minimal tuning. We think that it is reasonable to assume that the gap will increase by optimizing the hyper-parameters
>
> > "More experiments will be needed to justify"
>
> We have added results using the ConvNeXt backbone across four datasets and three model sizes. Please refer to the section '2-D SSM as an Alternative to Standard Convolution Layers (Experiments with ConvNeXt)' in the shared response, and Tables 6 and 7 in the revised manuscript for detailed information.
>
> > “Besides image level classification, can you run experiments on pixel level tasks such as semantic segmentation or instance segmentation by plug in this new layer into, e.g., Mask2Former or Pyramid Vision Transformer?”
>
> Traditionally, such experiments rely on large pre-trained models (trained on ImageNet-22\1K or through SSL schemes), and therefore are very costly and out of our budget’s reach. Please refer to 'Nature of our Benchmarks' in the main response for more details. We acknowledge the value of such experiments; however, we believe that our extensive evaluation across six datasets, with several backbones (ConvNeXt, ViT, Swin, Mega, DeiT), and a variety of model sizes—from DeiT/Mega-Base (85M params) and Swin-T (27M params) to ConvNeXt-Micro (9M params)—provides a sufficient assessment to justify the empirical contribution of our 2-D SSM layer.

---

### Author Response · Authors · 2023-11-19
**Main Response by Authors**

We would like to thank the reviewers for their thoughtful comments and constructive feedback.

Following the reviews, we have uploaded a revised version of the manuscript which contains the following material (modifications in red):

1.  **2-D SSM as an alternative to standard Convolution layers (Experiments with ConvNeXt):**

In response to Reviewer ZC1E's suggestion, we assessed our layer as a Conv2D alternative within the ConvNeXt architecture. The revised manuscript includes the results of this evaluation in Tables 6 and 7, showcasing consistent performance gains across various datasets and model sizes. Notably, with the ConvNeXt-T model on ImageNet-100, the integration of 2-D SSM layers led to a 1.09% increase in performance and a reduction in parameter count. The ConvNeXt-M model saw an even more substantial improvement of 1.3%. We also observed a performance boost of 0.5-2% on the Tiny ImageNet, Celeb-A and CIFAR-100 datasets with our variants. It is important to emphasize that these enhancements were attained without hyper-parameter tuning, using configurations optimized for the original ConvNeXt model. Further insights into our methodology are detailed in Appendix G (Experimental Setup for ConvNeXt). Considering the widespread use of ConvNeXt as a solid benchmark across various research domains, we believe these results hold significant promise.

2. **Latency and Memory analysis:**

 In response to Reviewers ZC1E, mKTe, and SZ7R, we further investigated the latency and memory consumption of our layer. By profiling the training time to understand the current bottlenecks, we enhanced the layer by modifying the structure of the cached matrices and grouping the FFT computations for different directions. This optimization resulted in a 30% decrease in inference time and a 25% decrease in training time (on both A100 and RTX-4090).

Second, we wish to highlight that, unlike our approach with the ViT experiments, we did not add additional components in the ConvNeXt experiments, instead, we replaced Conv2D layers with 2-D SSM layers, which resulted in a smaller increase in training time. For instance, in experiments with ConvNeXt-N, our training time is only 1.26 times longer than the standard CUDA-dedicated and highly-optimized implementation of Conv2D-based ConvNeXt.

It is important to note that from a theoretical perspective, the 2-D SSM layer is a relatively efficient layer with sub-quadratic complexity in sequence length. Its structure allows for full parallelism of the forward and backward operations, thanks to our caching mechanism. Additionally, our layer is efficient in large-batch regimes, as the computation of the kernel does not involve the batch dimension.

Furthermore, in response to reviewer mKTe, we have included a detailed report on the actual inference times and memory footprint in Appendix H of the revised manuscript. Figure 8 demonstrates that our model is more memory-efficient compared to transformers, especially for large sequences. Figure 7 illustrates that the inference time of the 2-D SSM layer is faster than that of Transformers for any sequence length. Figure 6 analyzes the contribution of each model component to the total inference time of our method for ViT. We believe this section offers a comprehensive view of the 2-D SSM layer's performance.



3.  **Nature of our benchmarks:**

 we would like to address multiple concerns regarding the nature of our benchmarks. Reviewers CN5B and SZ7R suggest performing pixel-level tasks such as semantic segmentation using methods like Mask2Former or Pyramid Vision Transformer, which require costly pre-training on ImageNet-1\22K. Reviewer ZC1E suggests assessing the method after large-scale pre-training, and Reviewer mKTe argues that training on datasets like IN1K with 'Base' or larger models has become the norm in the domain. We recognize the value of these experiments. However, our limited resources prevent us from training such models from scratch on ImageNet-1K. Considering the published numbers for MEGA, as a comparable model, such an experiment would cost us north of 15,000$.

To address these concerns, we conducted experiments on ImageNet-100, the largest dataset our budget can accommodate. Since we use the same hyperparameters and training procedures (training duration, data augmentation, regularization) and achieve state-of-the-art results on this dataset, we believe it is reasonable to assume that with the required computational resources, the 2-D SSM layer has the potential to provide state-of-the-art results on large scale benchmarks as well (such as Imagenet-1K).

---

### Meta-Review · Area_Chair_yAbM · 2023-12-06

**Metareview:**

The paper introduces a two-dimensional recurrent state space model (2D SSM) layer for Vision Transformers (ViT), aiming to improve classification accuracy with a focus on incorporating spatial inductive biases in neural network architectures.  All four reviewers rated the paper as marginally above the acceptance threshold. They collectively recognize the theoretical innovation and potential practical applications of the proposed 2D SSM layer. However, they also agree that the paper has limitations in terms of computational efficiency, scope of application, and the need for more comprehensive evaluations across different models and tasks. The AC agrees with the reviewers to accept the paper, but the authors should well address the issues raised by the reviewers in the final version.

**Justification For Why Not Higher Score:**

The reviewers agree that the paper has limitations in terms of computational efficiency, scope of application, and the need for more comprehensive evaluations across different models and tasks. The concerns about the marginal improvement and potential instability under different conditions suggest that further refinement and extensive testing of the proposed method may be necessary to fully realize its potential in the field of computer vision.

**Justification For Why Not Lower Score:**

All four reviewers rated the paper as marginally above the acceptance threshold. They collectively recognize the theoretical innovation and potential practical applications of the proposed 2D SSM layer.

---

### Decision · Program_Chairs · 2024-01-16

Accept (poster)